# An Assay System to Evaluate Riboflavin/UV-A Corneal Phototherapy Efficacy in a Porcine Corneal Organ Culture Model

**DOI:** 10.3390/ani10040730

**Published:** 2020-04-23

**Authors:** Anna Perazzi, Chiara Gomiero, Livio Corain, Ilaria Iacopetti, Enrico Grisan, Marco Lombardo, Giuseppe Lombardo, Gianni Salvalaio, Roberta Contin, Marco Patruno, Tiziana Martinello, Antonella Peruffo

**Affiliations:** 1Department of Animal Medicine, Production and Health, University of Padova, 35020 Legnaro, Italy; ilaria.iacopetti@unipd.it (I.I.); skirobi93@gmail.com (R.C.); 2Department of Comparative Biomedicine and Food Science, University of Padova, 35020 Legnaro, Italy; chiara.gomiero@studenti.unipd.it (C.G.); marco.pat@unipd.it (M.P.); tiziana.martinello@gmail.com (T.M.); antonella.peruffo@unipd.it (A.P.); 3Department of Management and Engineering, University of Padova (PD), 36100 Vicenza, Italy; livio.corain@unipd.it; 4Department of Information Engineering, University of Padova, 35100 Padova, Italy; enrico.grisan@unipd.it; 5School of Engneering, London South Bank University, London SE1 0AA, UK; 6Vision Engineering Italy srl, 00198 Roma, Italy; mlombardo@visioeng.it; 7Consiglio Nazionale delle Ricerche, Istituto per i Processi Chimico-Fisici (CNR-IPCF), 98158 Messina, Italy; giuseppe.lombardo@cnr.it; 8Fondazione Banca degli Occhi del Veneto (FBOV), 30174 Venezia Zelarino, Italy; gianni.salvalaio@fbov.it

**Keywords:** riboflavin/UV-A corneal phototherapy, porcine cornea, image analysis, animal model, melting ulcers

## Abstract

**Simple summary:**

The scope of this study is to quantitatively evaluate, with an automated digital image analysis method, the efficacy of riboflavin/UV-A corneal phototherapy on the cornea in a porcine corneal organ culture model of ulcerative melting keratitis. Riboflavin/UV-A corneal phototherapy provided a favorable outcome in the corneal wound healing process after chemical injury: the treatment restores the damaged corneas to the texture of healthy corneas. This automated image analysis method may be compared to clinical diagnostic methods, such as optical coherence tomography (OCT) imaging, for in vivo damaged ocular structural investigations. Positive results from this research could provide an opportunity for studying the effects of this method in other economically and emotionally valued species, such as dogs, cats, and horses. The relatively overall low treatment cost and the ease of performing the procedure make riboflavin/UV-A corneal phototherapy accessible to the veterinary market.

**Abstract:**

The purpose of this study was to investigate the response of porcine corneal organ cultures to riboflavin/UV-A phototherapy in the injury healing of induced lesions. A porcine corneal organ culture model was established. Corneal alterations in the stroma were evaluated using an assay system, based on an automated image analysis method able to (i) localize the holes and gaps within the stroma and (ii) measure the brightness values in these patches. The analysis has been performed by dividing the corneal section in 24 regions of interest (ROIs) and integrating the data analysis with a “multi-aspect approach.” Three group of corneas were analyzed: healthy, injured, and injured-and-treated. Our study revealed a significant effect of the riboflavin/UV-A phototherapy in the injury healing of porcine corneas after induced lesions. The injured corneas had significant differences of brightness values in comparison to treated (*p* < 0.00) and healthy (*p* < 0.001) corneas, whereas the treated and healthy corneas showed no significant difference (*p* = 0.995). Riboflavin/UV-A phototherapy shows a significant effect in restoring the brightness values of damaged corneas to the values of healthy corneas, suggesting treatment restores the injury healing of corneas after lesions. Our assay system may be compared to clinical diagnostic methods, such as optical coherence tomography (OCT) imaging, for in vivo damaged ocular structure investigations.

## 1. Introduction

The cornea is the outmost layer of the eye and represents the most important site for appropriate focusing of the image to the retina. Any trauma or disease that affects the cornea may have a significant impact on vision. A corneal ulcer is a potentially sight-threatening process; it may present with keratomalacia, often termed a “melting ulcer.” The term “melting” describes the rapid gelatinization and liquefaction of the corneal stroma, as a result of an imbalance between proteinases and proteinase inhibitors, and loss of rigidity and structure of the corneal collagen and matrix [1,2,3,4].

Although melting ulcers may respond to medical management, a progression of stromal dissolution and corneal edema is frequent, serious, and rapidly progressive, and surgical treatment is necessary to avoid corneal perforation [3,5,6,7]. In recent years, research efforts, both in human and veterinary medicine, have been directed in the search for alternative or complementary methods that could support the healing processes of corneal ulcerative diseases [3,7,8]. Particular interest has been directed towards a method designed to strengthen the structure of the corneal stroma, working on the natural links between collagen fibers, to avoid graft and corneal transplantation.

UV-A/riboflavin corneal cross-linking (CXL) was introduced as a treatment for humans in 1999 and is currently used to treat primary corneal ectasia, such as progressive keratoconus, and iatrogenic corneal ectatic disorders [4,6,7,9,10,11,12]. The technique works through the application of riboflavin (vitamin B2) as a photosensitizer and UV-A irradiation of the cornea with the aim of strengthening the corneal biomechanics due to the increased formation of covalent bonds between the stromal proteins by photopolymerization and photooxidation processes [8]. These bonds increase collagen fiber resistance and stiffness, resulting in more corneal stability and prevention of the progression of corneal ectasia [13,14].

During the last five years, several authors in both human and veterinary medicine have assessed efficacy of riboflavin/UV-A corneal phototherapy as an adjuvant treatment in cases where medical therapy had failed to control infectious melting keratitis [2,4,7,9,14,15,16,17,18,19,20,21,22,23,24,25]. The term riboflavin/UV-A corneal phototherapy is used to distinguish the use of the technique for the treatment of keratitis from its primary indications for ectatic disorders [20].

Studies of ex vivo corneas have been performed using the healthy porcine cornea [26,27], and in horses [4], rabbits [4] and humans [28], to evaluate the efficacy of CXL treatment. However, as far as the authors know, few controlled studies of porcine corneal organ culture models aiming at assessing the efficacy of riboflavin/UV-A corneal phototherapy for induced melting keratitis have been undertaken [29,30].

The scope of this study was to quantitatively evaluate the efficacy of riboflavin/UV-A corneal phototherapy on the porcine cornea in a porcine corneal organ culture model of induced ulcerative melting keratitis. A novel automated method for digital image analysis was designed to measure the structural alteration in the stroma, quantifying the stromal inhomogeneity (“damaged fissures”) using the brightness in the cornea section as a proxy.

## 2. Materials and Methods

### 2.1. Eyes Preparation

Porcine eyes were obtained from local abattoirs immediately after slaughter. All animals were treated according to Italian and European (86/609/EEC) regulations concerning animal welfare during the commercial slaughtering process and were constantly monitored under mandatory official veterinary medical care. Enucleation was performed immediately after death. The eyes were transported in a 10% povidone-iodine solution and transported chilled at 4 °C to the laboratory. The whole eyes were washed in phosphate-buffered saline (PBS) (Dulbecco’s PBS; PAA Laboratories) for 5 min and then processed according to the respective treatment group working in sterile conditions (Figure 1a). All corneas were found to be clear, with no presence of corneal scarring or opacities. No corneal pathologies were found.

### 2.2. Experimental Groups

In total, 30 corneas were analyzed in this experiment. To investigate the response to the riboflavin/UV-A phototherapy, porcine corneas were randomly divided into 3 groups: the healthy group, (*n* = 10 corneas as control), the injured group (*n* = 10 corneas that received a chemical injury), and the treated group (*n* = 10 corneas that underwent riboflavin/UV-A corneal phototherapy after the induced chemical injury).

### 2.3. Induction of Experimental Lesion and Riboflavin/UV-A Corneal Phototherapy Procedures

The experimental lesion was induced with a chemical technique to obtain an alkali-induced corneal stromal melting, as reported in rabbits [31,32,33,34]. The induced lesions on the cornea were produced using a filter paper (0.8 cm) soaked with NaOH (1N) that caused significant tissue damage on contact. The filter paper was placed for 1 min on the center of the ocular surface (Figure 1b); then the corneas were washed with PBS solution for 60 s. The group of injured corneas did not receive any treatment, whereas the corneas injured in the treated group were treated with corneal phototherapy. Isoosmolar 0.1% riboflavin drops (Peschke Traid, Huenenberg, Switzerland) were administered into a circular plastic well held firmly against the cornea for 30 min to prevent excessive fluid drain and assure riboflavin saturation of corneal stroma (Figure 1c). The corneas were then irradiated for 3 min with commercially available equipment (Vetuvir™, Vision Engineering Italy srl, Rome, Italy) following the manufacturer’s instructions (Figure 1d). The wavelength was 365 nm and the irradiance was 30 mW/cm^2^ (for total UV-A energy of 5.4 J/cm^2^). The diameter of irradiation was 9 mm. The UV light was focused on the corneal surface at a distance of 10 cm. At the end of the treatment, the ocular surface was thoroughly irrigated with PBS solution for 60 s.

### 2.4. Porcine Organ-Culture Procedure

After the experimental procedures described above, the corneas were isolated from the bulbs, and the corneal-scleral rims, with approximately 4 mm of the limbal conjunctiva present, were excised. The isolated corneas were suspended into 25 mL sterile bottles using suture thread passed through the sclera and preserved in a specific culture medium for corneal deturgescence (Carry-C^®^ medium, Alchimia, Padua, Italy) for 7 days in an incubator at 37 °C with 5% CO_2_.

At the end of the culture period, the corneal endothelium was examined by light microscopy before and after culture. We counted the number of dead cells by staining the corneas with trypan-blue. After staining, the corneas were rinsed with phosphate buffered saline (PBS). We found a loss of endothelial cells from 10%–20%, enabling preservation of porcine corneas for at least 7 days.

### 2.5. Histological Staining

After 7 days in culture, the porcine corneas were fixed in 10% formalin and embedded in paraffin for 10 days, and then processed for paraffin embedding. Serial cuts were performed at regular intervals of 1000 μm (Figure 2) to examine each cornea in the totality of its diameter; the protocol provided the elimination of a first portion of the sample that was considered not significant for the evaluation of the lesion area. From each cornea, transverse sections (slides) of 5 μm thickness were obtained at every cutting point using a Leica RM2035 microtome. The slides were stained with Hematoxylin and Eosin (H&E) according to the manufacturer’s instructions following a routine protocol [35]. Sections were then dehydrated in xylene and finally, cover-slipped with a mounting medium for cover slippers (Entellan Sigma-Aldrich) for microscopic evaluation.

Scanning of stained sections was performed with a semi-automated microscope (D-Sight v2, Menarini Diagnostics, Italy) at a magnification of 20× in fast mode. The images were exported as JPeg2000 files to be processed by image analysis. An assessment of corneal quality was performed independently by three different observers (AP, AP, II) checking 2–4 H&E-stained sections of each cornea using at least 20× magnification. The evaluation included parameters, such as the stroma collagen compaction, and the number and distribution of cell nuclei in the stromal thickness in the endothelium and in the epithelium.

### 2.6. Image Processing and Data Collecting

The complete analysis of the acquired images of corneal slices involved the detection of tens of thousands of fissures in the stroma. This is not feasible by human annotation of the images, unless the procedure is carried out for a small region of interest, potentially introducing bias in the procedure.

To tackle this problem, and assuming that the area covered by holes and gaps is correlated with the collagen damage, we developed an automatic procedure that can process the images and quantify the damage through the localization of holes and gaps in corneal tissue. This method identifies the outline of the stromal damages (fissures in the tissue) and detects the brightness values as light intensity within the fissures among the different corneal populations (see Appendix A).

The question is how to understand image brightness; the brightness values are numerical information consisting of the grey values that describe the brightness of every point within the image. This information was used to analyze alterations of grey value intensities in the corneal stroma. We assumed that low values of brightness are linked to higher stromal compactness, while high values of brightness indicate greater stromal laxity. In this way, the healthy tissue shows a low average intensity of brightness, whereas damaged tissue shows high values of brightness.

In order to compare the corneal alterations in the stroma in the injured and treated corneas, we first assessed the brightness values of the fissures present in the thickness of the healthy corneas stained with H&E. The healthy brightness values were considered control values.

The healthy brightness values were evaluated in the healthy corneas in the 4 layers and in the 7 radial frames separately. To divide the corneal stroma into regions of interest (ROIs) allowed us to specifically observe the regional brightness variations.

To evaluate the alterations in the corneal stroma of healthy, injured, and treated porcine corneas, we developed an in-house pipeline for the automatic analysis of digitized images of the cornea sections, which was able to automatically characterize and measure the brightness values in the stroma of the corneal section (for details see Appendix A). We assumed that high values of brightness suggest stromal laxity in the cornea, while low brightness values suggest stroma compactness.

Since it is assumed that both the lesion and the treatment of the cornea have a local effect that spreads into the surrounding tissues in an unpredictable way, any measure evaluating a change in the tissue was performed as locally as possible. The benefit of dividing the corneal surface sections into layers, frames, and ROIs is specifically to display the regions showing statistically significant values of brightness within the corneal thickness. To obtain both a global assessment and the effect of treatment using local values of brightness, we chose to automatically divide each cornea section into 24 regions of interest, separating the section into 4 layers with boundaries parallel to its main axis, and 6 radial sections with boundaries perpendicular to the axis (Figure 3).

#### 2.6.1. ROI Detection

Synthetically, after considering the green channel of a stained image, a binary threshold was identified using the Otsu method [36], which separates the image histogram in such a way as to maximize the inter-class variance of the identified regions (one comprising all pixels with intensity lower than the threshold, the other all pixels with intensity higher than the threshold). After applying a set of morphological filters to remove spurious and isolated regions to the binarized images, the remaining region with the larger area was considered as the cornea (Figure 4a). The center line (main axis) of the identified region was then identified on the selected region with a method similar to that used by Grisan et al. [37]: a third-order polynomial fitted to the binary mask of the cornea was used as the starting estimate of the central line (Figure 4b). For each point belonging to this polynomial, a straight line passing through it and perpendicular to the line was analyzed, identifying the limit of the cornea region and updating each center point, computing the one along the line having the same distance from the opposite cornea boundaries (Figure 4c). A cubic smoothing spline was then fitted to the updated center points to obtain a regular and equally sampled center line (Figure 4d). The estimated central line allowed the subdivision of the cornea section into 4 layers and 6 radial frames (Figure 3). The first layer (outer) included the epithelium and the most anterior stroma of the Bowman layer acellular, and a part of the stroma; the second and third layers included the central area of the stroma containing collagen fibrils; and the fourth layer included the posterior part of the stroma and the inner endothelium.

#### 2.6.2. Collagen Damage Quantification

The accurate evaluation of collagen damage requires the detection of bright fibrils and of bright unstained patches indicating the presence of gaps and holes in the tissue. We assume that the area covered by these patches is correlated with the collagen damage.

However, the detection of these patches is tricky due to the non-uniform H&E staining, and to the possible presence of partial damage within the sample thickness (partial volume effect) that will go undetected (see Appendix A), thereby biasing the estimated damage.

Considering that healthy tissue should appear uniformly stained with small regular bright fibrils in H&E images, whereas damaged corneas should show larger gaps and holes in the tissue that appears as bright (white) spots in H&E images, we would like to provide a more robust and sensitive measure of the ongoing (and progressive) changes in the corneal tissue than the one obtained with a localization approach.

In order to achieve this, we chose to use the simple average intensity of the green channel of each region as a proxy for the quantification of collagen damage. *I_g_* is the green channel of an H&E image; *R_ij_* is one of the regions identified during the preprocessing (see Section 2.6.1) with corresponding area *A_ij_*; *i* = 1, …,4 is the layer index of the region and *j* = 1,…,6 is the radial position. The brightness measure is computes as:
$$b_{ij}=\frac{1}{A_{ij}}\int_{R_{ij}}I_{g}\;dR=\frac{1}{A_{ij}}\sum_{\left(r,c\right)\in R_{ij}}I_{g}\left(r,c\right)$$


Assuming a fairly uniform staining appearance (from the presence of normal tissue and fibrils) across healthy samples, those measures will provide an average baseline intensity (brightness) of normal tissue. In presence of an increased quantity of collagen damage, an increased total area of unstained (bright) regions will appear and will be reflected on the measure, providing a higher value than the healthy baseline. Additionally, since the measure does not need hard detection of damage areas, but rather evaluate the changes in stain uptake by the tissue, it will not be affected by partial volume artifact, as a small local increase in image intensity will contribute to the regional average.

By those means, low values of average brightness measure are linked to higher compactness, whereas high values of average brightness indicate greater laxity.

### 2.7. Statistical Design and Data Analytics

The focus of data analytics was the comparison among the three populations (corneal groups under investigation). A multi-way ANOVA [38] was used to prove the causal relationship between a set of input factors; i.e., the population and the specific ROI, versus a numerical response; i.e., the brightness we measured as a proxy of the damaged tissue (see Section 2.6). In order to prove possible significant effects due to the population and the ROI, we also applied nonparametric permutation tests, formerly already considered in image analysis with either fMRI [39,40] or histological data [41,42]. This methodology can be considered the more recommended statistical approach to our histological data, because of their possible non-normal distribution [43,44]. We also applied specific innovative multi-aspect tests to provide additional insights on the comparison among populations. Details of these latter tests are presented as Appendix A—statistical design and data analytics. Briefly, this methodology consists of a powerful nonparametric approach [45] able to quantify fine differences in the tissue. With the term “multi-aspect,” we mean that we are focusing on two different distributional aspects of the histological measure; i.e., the location (the mean) and the scatter (the variance). Results of pairwise testing were then exploited to apply the multivariate ranking methodology recently proposed by several authors [46,47,48]. The multi-way ANOVA F-test method was used to analyze the possible synergic effects between population, depth layer, and radial frame (the latter are the two factors that define the relative position of the ROI). More details can be found in the Appendix A.

For all tests, a *p*-value of less than 0.05 was considered to be significant.

## 3. Results

### 3.1. Corneal Organ Culture Results

All corneas remained clear and did not develop edema or infection over the period of 7 days in cultures. The living corneas in cultures were examined daily using a phase-contrast microscope to monitor the epithelium, the endothelium, and the stromal tissue quality. The endothelial cells’ morphology and vitality did not differ during the 7 days. Corneal thickness did not show any statistically significant difference during the 7 days. The swelling of the cornea preserved in the culture medium was weakly present or absent.

### 3.2. Histological Evaluation of Hematoxylin and Eosin-Stained Sections

In the population of healthy corneas, the stroma showed compact bundles of lamellae in a parallel manner showing a high cellularity, and no shrinkage was detected throughout the thickness (Figure 5A). In the population of healthy corneas, the Bowman layer (the most anterior stroma) appeared acellular, and was composed of oriented and interwoven collagen fibers. The endothelium was intact, showing a single layer of hexagonal cells with a stained nucleus.

The population of injured corneas showed an important alteration in the structure of the corneal stroma. The bundles of collagen fibers appeared unstructured, showing numerous fissures in the damaged area of variable dimensions. The endothelium appeared intact and cellularized (Figure 5B). The population of treated corneas after injury showed more organized and compact collagen lamellae with respect to injured corneas. Furthermore, the fissures between the lamellae appeared to be reduced compared to the untreated corneas, which showed a higher cell density close to the endothelium (Figure 5C).

### 3.3. Statistical Analysis

The values of brightness collected by image analysis of the three corneal populations were statistically analyzed and the descriptive results (by boxplot) were supported by inferential analysis through pairwise comparisons to specifically evaluate the differences between pairs of populations (healthy vs. injured, healthy vs. treated, injured vs. treated).

### 3.4. Statistical Analysis of Brightness Values among Corneal Populations

As a first step we analyzed the brightness values by boxplot representations among the healthy, injured, and treated corneal populations (Figure 6a). Comparing the boxes of the healthy corneas vs the boxes of the injured corneas, it is possible to note that the boxes of injured corneas show an upward shift, indicating that injured corneas have higher brightness values (Figure 6a). The boxes of treated corneas by phototherapy shows brightness values lower than the boxes of injured corneas, suggesting a reduction of the damage in the stroma in this population (Figure 6a).

This evidence is supported by the fact that both the sample means and medians of brightness values from healthy and treated corneas are close each other, while the injured corneas group has the largest mean and median values (Table 1).

To highlight statistical significance between corneal groups, the means and SDs were calculated separately for the four layers (layers 1, 2, 3, and 4). This significance is due to layers 2 and 3 (95% confidence interval), as shown in Figure 7 and the corresponding Table 2.

The inferential analysis among the three corneal populations confirmed the significant effect (*p*-value < 0.001) of populations in brightness values (Table 3).

Results of post-hoc pairwise comparisons between corneal populations showed that injured corneas had a larger brightness mean value than that of both the healthy and treated corneas (both *p* < 0.001), while the comparison between healthy and treated populations was not significant (*p* = 0.995) (Table 4). In detail, the treated corneas showed a brightness mean value that was similar to that of the healthy corneas.

This result suggests that the extensions of the holes and gaps in the injured corneas was significantly greater than in both healthy and treated corneas, in which extension had smaller values. As suggested by the values of all variance-related statistics by corneal population (Table 1), there was a more scattered distribution of the brightness values in the population of treated corneas in comparison to both injured and healthy groups (*p* < 0.001).

From analyzing the means (location) and variances (scatter) of the brightness among the three corneal populations from the smallest to the largest values (for details see the Appendix A—statistical design and data analytics), the results confirmed that the healthy and treated corneas are equal in mean and both have a lower value than for the injured corneas (Table 5). Moreover, the treated corneas are the population with the largest variability in the brightness values (Table 5).

### 3.5. Statistical Analysis of Brightness Values Distribution among Layers and Radial Frames

We studied the brightness values distribution within the four corneal layers (Figure 6b and Figure 8) and in the six radial frames separately (Figure 6c and Figure 8).

Concerning the distribution of the brightness value in the four layers, it is possible to observe that the box of layer 1 and the box of layer 4 are positioned lower than the boxes of layers 2 and 3 and this condition is maintained in each of the three corneal populations (Figure 5B).

These results indicate that within the corneal thickness, the two central layers possess higher brightness values than the two outer layers, suggesting a lower presence in this area of holes and gaps, independently of the treatment or condition of the cornea (Figure 5B). This result suggests that the first (outer) and the fourth (inner) layer analyzed show a greater compactness in comparison to the second and third layer (all *p* < 0.001).

The inferential analysis among layers and radial frames revealed a significant effect in the case of layers (*p* < 0.001), while inferential analysis among radial frames showed no significant effect (*p* = 0.234) (Table 3).

### 3.6. Statistical Analysis of Brightness Values Distribution among the 24 ROIs

Finally, we analyzed for each of the 24 ROIs (four layers and six radial frames) the permutation *p*-values between healthy vs. treated corneas. By the matrix heat map, we graphically represent the comparison between healthy and treated corneas (Figure 9). The equality in location (mean of brightness values) between treated and healthy corneas is confirmed in each of the 24 ROIs (*p* > 5%).

It is interesting to note that the differences in the scattered distributions of the brightness values that we previously found in the population of treated corneas in comparison to both injured and healthy corneas (*p* < 0.001) are not equally spread across all ROIs (Figure 9). In particular, the ROIs in which the *p*-values are the smallest are present in the fourth radial frame and in the fourth layer.

This result in the fourth radial frame could be due to higher histological damages caused during the experimental procedure to induce the lesion; the fourth radial frame indeed represents the most exposed frame to soda treatment compared to the lateral radial frames.

In addition, the fourth layer is the layer with the lowest *p*-value. This result could be due to the minor efficiency of riboflavin/UV-A phototherapy in terms of its ability to penetrate the inner layer compared to the other layers.

## 4. Discussion

In this study, we investigated the riboflavin/UV-A phototherapy response of porcine organ cultured corneas in injury healing after induced lesions, developed an automated image analysis method to quantify the stromal damage (brightness values), and analyzed the data with a multi-aspect approach.

Our results showed that injured corneas had a larger mean brightness value than both the healthy and treated corneas, while treated corneas had a mean brightness value that was equal to the value of the healthy corneas. Interestingly, the outer layer and the inner layer per cornea section showed lower brightness values than the two central layers in each of the three corneal populations.

The porcine cornea organ culture model offers a standardized model for assessing responses in the mechanisms of repair process of riboflavin/UV-A phototherapy in ulcerative melting infectious keratitis in a dynamic system. Moreover, it represents an alternative approach in response to ethical concerns increasing demand to replace the current in vivo methods.

We maintained the porcine corneas in culture for a fixed time of 7 days in order to keep corneas healthy and to avoid uncontrollable variables, thereby affecting the treatment response due to maintenance in the culture for a longer time.

The presumed mechanisms of the riboflavin/UV-A corneal phototherapy repair process involve an increase in collagen packing density and a reduction in the swelling tendency of the glycosaminoglycan-rich hydrophilic ground substance of the cornea [21]. In this way we assumed that high values of brightness correspond to greater damage, suggesting stromal laxity in the cornea, while low brightness values suggest stroma compactness. This assumption allowed us to quantify and analyze the results of brightness values among healthy, injured, and treated cornea populations.

Despite strong evidence published by several groups [2,4,7,9,14,15,16,17,18,19,20,21,22,23,24,25] indicating the beneficial effect of riboflavin/UV-A corneal phototherapy, one problem for researchers is the lack of automated methodologies with which to characterize the repair process mechanisms throughout the corneal thickness. Additionally, manual interpretation could be prone to inter-observer variability.

With the aim to overcome that shortcoming, we set up an assay system able to automatically measure the corneal damage (brightness values) in the thickness of cornea sections, integrating this analysis with a multi-aspect statistical approach of data analytics. Our results indicate that treated corneas showed brightness values equal in mean to the values of the healthy corneas, leading us to speculate that there is a possible positive effect of the riboflavin/UV-A corneal phototherapy in the repair process of corneas after induced lesions. In addition, the interesting result that showed the treated population had a more scattered brightness distribution than both the healthy and injured populations, letting us hypothesize a different ability of riboflavin/UV-A phototherapy to reduce the extension of white fissures in relation to the extension of the damaged surface, suggesting a complete regeneration of the stromal structure when the injury area is small, and a loss of regenerative capacity or an incomplete repair process when the injury area is larger.

It must be considered that in a mammal’s cornea, collagen organization of the anterior stroma has a very different arrangement when compared with the posterior cornea. As specified by Leonard and colleagues, there is a direct correlation between corneal collagen stromal organization and tissue stiffness in the mammalian species, and this likely reflects the need for maintenance of rigidity and corneal curvature [50].

In our study, these differences were also highlighted by analyzing the brightness value distribution within the four corneal layers (as described by the main effects plot for depth layer in Figure 8), and this condition was maintained in each of the three corneal populations.

Results showing that outer layers have lower brightness values, in comparison to the two central layers, independently of the treatment or condition of the cornea, indicate a greater compactness of the outer layers in comparison to the central layers in each of the three cornea populations analyzed. A structure that presents external compactness enclosing a central laxity gains a well described property named the “sandwich effect” [51]. The sandwich effect encloses the property postulated from the well-known sandwich theory that describes the behavior of a structure consisting of three layers: two face sheets and one core. This assembly confers the property of minimizing resistance and increasing plasticity. This could be an intrinsic property of the cornea structure that permits the cornea to support and minimize resistance due to internal pressures.

This is the first time that riboflavin/UV-A phototherapy has been analyzed in distinct topographical corneal regions (ROIs). The analysis of brightness values, performed in distinct ROIs, enables relating the results to a precise spatial position within the cornea section. Moreover, the graphical representation by heat map allows the visualization of the distribution of brightness results, relating brightness data to a precise spatial position within the cornea section.

As pointed out by Poldrack [52], ROI-oriented image analysis can provide substantial insights into the nature of effects in complex models, as well as provide valuable assistance in diagnosing model failures. By using these focal methods, and subdividing the thicknesses of the cornea in ROIs, it is possible to identify the histological areas in the corneal stroma where the brightness values gave statistical significance. Moreover, the subdivision of the thickness of the cornea in ROIs allows for visualization of specific hidden effects in the analysis conducted between layers and between frames. These methods may be used in future as a base for obtaining new insights into normal corneal stroma, and consequently, to also determine quantitatively the presence of corneal disease. Nonetheless, it is worth noting that the ROI analysis we presented as a heat map-like figure (in Figure 9) must be considered as a preliminary exploratory tool.

While there is a need to have automated scoring mechanisms to evaluate histologic data, the scientific approach taken in this study has some challenges and limitations. Thus, new methodologies that promote interdisciplinary collaboration are required to overcome potential obstacles.

## 5. Conclusions

The use of our assay system to evaluate riboflavin/UV-A phototherapy of porcine corneas provided encouraging outcomes in demonstrating the corneal wound-healing process after chemical injury. This methodology could be a helpful starting instrument for carrying out histological analysis in the field of corneal pathologies, in which knowing the extent of histological damage in a specific corneal layers or regions could be used for the assessment of the effect and the penetration depth of the treatment.

Finally, the encouraging results from this research could provide a new way to study the effects of riboflavin/UV-A phototherapy in other species of high emotional and economical value, such as dogs, cats, and horses.

## Figures and Tables

**Figure 1 animals-10-00730-f001:**
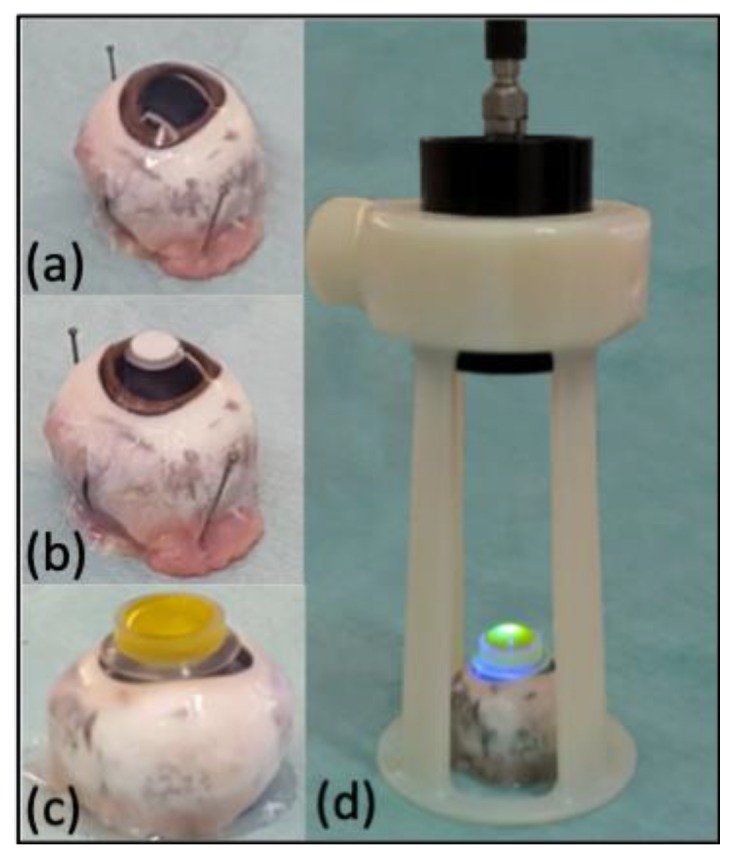
(**a**) healthy enucleated porcine eyes; (**b**) porcine eyes showing the filter paper placed on the center of the ocular surface; (**c**) application of isoosmolar 0.1% riboflavin drops into a circular plastic for 30 min; (**d**) details of the UV-A lamp with an irradiance of 30 mW/cm^2^ during riboflavin/UV-A corneal phototherapy.

**Figure 2 animals-10-00730-f002:**
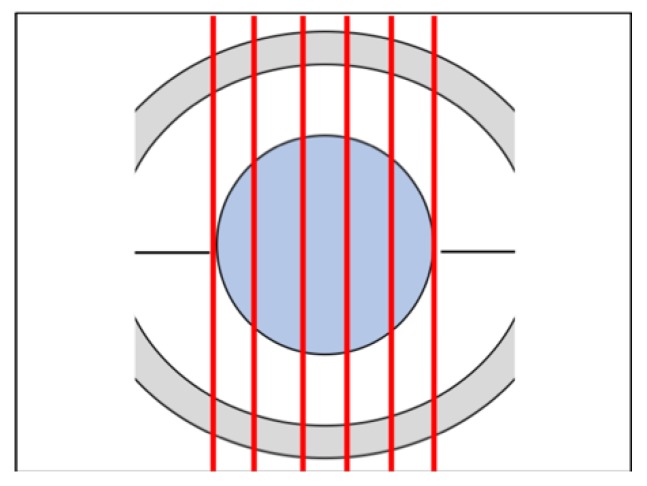
Schematic representation of the cornea, dorsal view; the central area (blue), identifies the area of the induced lesion in the cornea; the red lines represent the points for the collection of the histological slides. Serial cuts have been performed at regular intervals of 1000 μm from each (red lines) to examine each cornea in the totality of its diameter: the protocol provided the elimination of a first portion of the sample considered not significant for the evaluation of the lesion area. From each cornea, transverse sections (slides) of 5 μm thickness were obtained at every cutting point using a Leica RM2035 microtome.

**Figure 3 animals-10-00730-f003:**
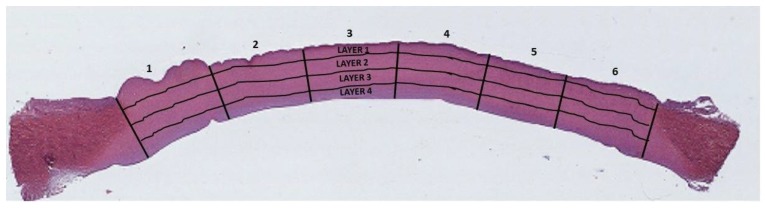
Cross-section of hematoxylin-eosin stained porcine cornea. The black lines indicate the regions into which the corneas are subdivided for performing the measurements. The grid outline of the 6 radial frames along the minor axis and the 4 layers along the major axis, thus obtaining 24 ROIs per corneal slides. The entire stromal thickness has been analyzed in each section of the three corneal populations considered. The stroma located at the top of the corneal section represents the anterior part of the cornea; the inferior part represents the inner (posterior) stroma of the cornea.

**Figure 4 animals-10-00730-f004:**
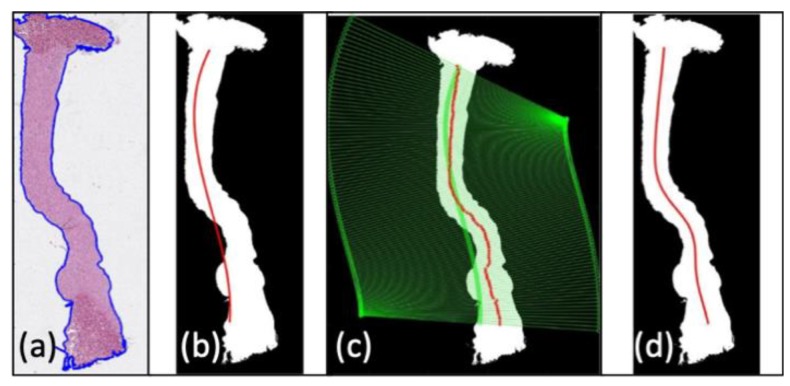
Automatic procedure for the identification of the cornea section boundaries and center-line; (**a**) section boundaries after thresholding and morphological filtering; (**b**) initialization of the center line with a polynomial (red); (**c**) analysis of straight lines (green) normal to the polynomial and update of the center points (red circles); (**d**) the final estimate of the center line. The stroma located on the right of the corneal section represents the anterior part of the cornea; the stroma located on the left side represents the inner (posterior) stroma.

**Figure 5 animals-10-00730-f005:**

Corneal section images after Hematoxylin & Eosin histological staining; (**A**) healthy cornea slide, note normal appearance of corneas’ thickness and the stroma show compact collagen lamellae; (**B**) injured cornea showing the alteration in the structure of the stroma, the lamellas of collagen fibers appeared disorganized with the presence numerous fissures; (**C**) details of the riboflavin/UV-A corneal phototherapy treated cornea, showing a more compact collagen lamellae respect to injured corneas. The fissures between the lamellae appear reduced. The stroma located on the right of the corneal section represents the anterior part of the cornea; the left part represents the inner (posterior) stroma of the cornea.

**Figure 6 animals-10-00730-f006:**
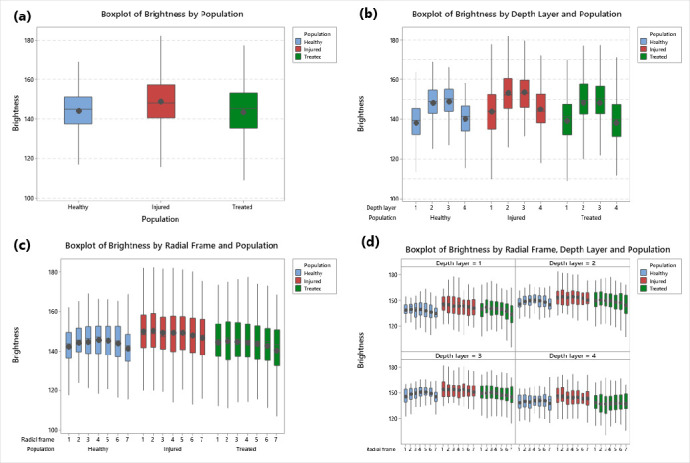
Boxplot representation of the brightness distribution among the corneal populations; (**a**) boxplot images comparing brightness by population; (**b**) by population and depth layer; (**c**) by population and radial frame; (**d**) by population and combination of the depth layer and radial frame.

**Figure 7 animals-10-00730-f007:**
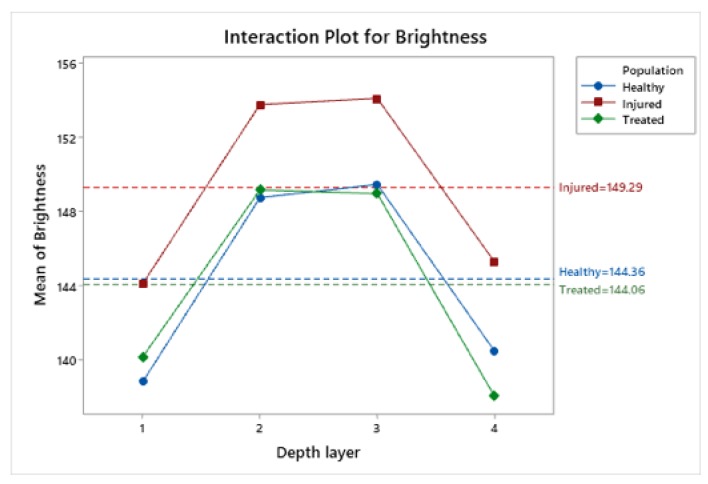
Interaction Plot showing the brightness distribution by population and the depth layer separately for the three corneal populations.

**Figure 8 animals-10-00730-f008:**
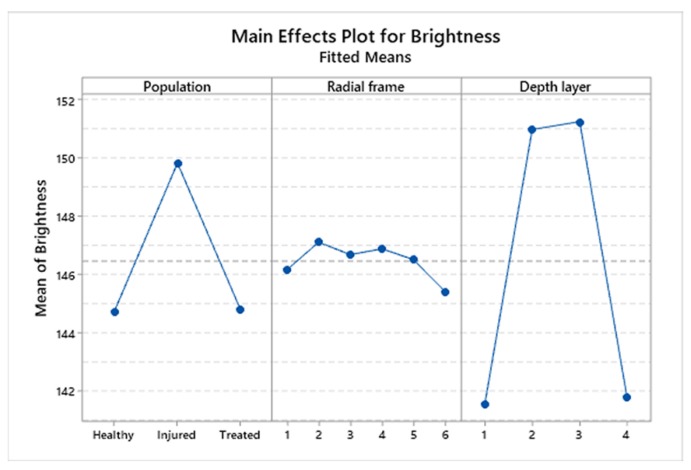
Main effects plot for brightness graphically representing the estimated fixed effect due to population, radial frame and depth layer.

**Figure 9 animals-10-00730-f009:**
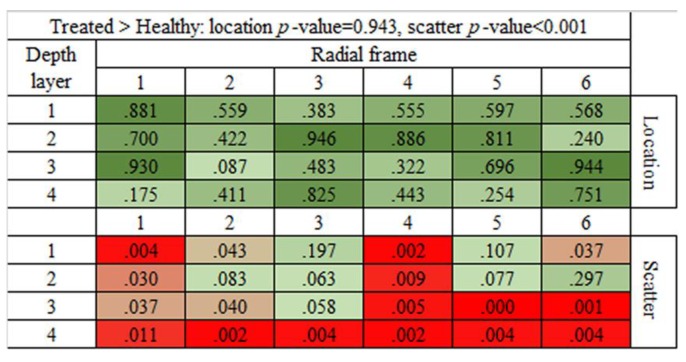
Heat map representing the 24 ROI (4 layers and 6 radial frames) of the cornea and the related *p*-values for the comparison treated vs. healthy corneas, (either in location and in scatter). The *p*-values < 5% are marked by a red color (more intense as the *p*-value gets smaller).

**Table 1 animals-10-00730-t001:** Sample mean, median, and variance-related statistics by corneal population (St.Dev: standard deviation; CV: coefficient of variation; IQR: interquartile range).

Population	Mean	Median	St.Dev	CV	IQR
Healthy	144.36	145.39	10.24	7.09	13.72
Injured	149.29	148.33	12.66	8.48	16.63
Treated	144.06	145.7	14.24	9.88	17.55

**Table 2 animals-10-00730-t002:** Mean, St.Dev (standard deviation), and 95% CI for each depth layer. The set of brightness values analyzed in the healthy corneal population were defined as threshold values for the control sample.

Population	Depth Layer	Mean	St.Dev	95% CI
Healthy		138.81	9.87	(137.54; 140.08)
Injured	1	144.08	12.71	(142.45; 145.72)
Treated		140.15	13.13	(138.56; 141.74)
Healthy		148.72	8.98	(147.57; 149.88)
Injured	2	153.74	11.64	(152.24; 155.23)
Treated		149.14	12.80	(147.59; 150.70)
Healthy		149.44	8.57	(148.33; 150.54)
Injured	3	154.07	11.20	(152.62; 155.51)
Treated		148.94	13.26	(147.33; 150.55)
Healthy		140.46	8.82	(139.33; 141.60)
Injured	4	145.26	11.61	(143.77; 146.75)
Treated		138.03	14.13	(136.32; 139.75)

**Table 3 animals-10-00730-t003:** Tests of fixed effects (either main effects and interactions) obtained by the normal-based mixed multiway ANOVA method. *p*-Values significant at 5% are highlighted in bold.

Term	F-Value	*p*-Value
Population	63.48	<**0.001**
Radial frame	1.37	0.234
Depth layer	166.28	<**0.001**
Population × Radial frame	0.93	0.505
Population × Depth layer	1.22	0.293
Radial frame × Depth layer	0.28	0.997

**Table 4 animals-10-00730-t004:** Post-hoc pairwise comparisons of the differences between means by the Tukey method. The 95% simultaneous confidence intervals (CI) and adjusted *p*-values significant at 5% are highlighted in bold.

Difference of Populations	Diff. of Means	95% CI	Adj. *p*-Value
Injured—Healthy	1503	**(1141; 1865)**	**<0.001**
Treated—Healthy	15	(−337; 367)	0.995
Treated—Injured	−1488	**(−1840; −1136)**	**<0.001**

**Table 5 animals-10-00730-t005:** Mean-based (test on location) and variance-based (test on scatter) permutation one-sided *p*-values for pairwise comparisons among populations and related ranking results. In each row the *p*-value refers to the comparison between the population in row vs. the population in column (hypotheses: population in row = population in col. vs. population in row > population in col.). The 1% significant *p*-values are highlighted in bold (since we jointly considered both one-sided alternatives, the actual α-level must be α/2). *p*-values were adjusted by multiplicity using the Bonferroni–Holm–Shaffer method [49].

Population	Mean			Variance		
	Healthy	Injured	Treated	Healthy	Injured	Treated
Healthy	-	1.000	0.070	-	1.000	1.000
Injured	**0.001**	-	**0.001**	**0.001**	-	1.000
Treated	0.931	1.000	-	**0.001**	1.000	-
*ranking*	2	1	2	3	2	1

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
