# Peer review of "An Assay System to Evaluate Riboflavin/UV-A Corneal Phototherapy Efficacy in a Porcine Corneal Organ Culture Model"

_animals, 2020, doi:10.3390/ani10040730_

Round 1
Reviewer 1 Report
You have replied satisfactorily to the various points requested.
Interesting research subject and good development.
Author Response
Just few words to say thank you very much.
Reviewer 2 Report
Thank you for your submission of an edited manuscript of the study entitled, "An assay system to evaluate riboflavin/UV-A corneal phototherapy efficacy in a porcine corneal organ culture model". While the edits have improved the manuscript, the major drawback of the study is the description and explanation of the "brightness" measure. There is no technique or technology described in the materials and methods section. While an automated analysis of corneal clefting may be important to determine the efficacy of a therapy, this manuscript lacks the clarity to evaluate its significance. The stromal clefting that are being identified as "brightness" is in the posterior corneal stroma. The supplementary figure shows this in greyscale, but the orientation of the cornea is not described, but based on the H&E images, it is shown upside down with the posterior stroma located at the top of the image.
Overall, it is hard to determine the utility of brightness in the context of this study. The concerns and recommendations of this reviewer are the same as in the original review.
Author Response
Please see the attachment.

This manuscript is a resubmission of an earlier submission. The following is a list of the peer review reports and author responses from that submission.
Round 1
Reviewer 1 Report
The subject of your article is well developed using different tools to highlight changes in the corneal stroma.
The automated images analysis method is very interesting and it allows a good correlation between injured/treated/ healthy corneas and brightness values.
I have only a few minors comments:
- Line 25 and 43: the verb “translated” should be replaced by the verb “compared”. The automated images analysis method may be compared to…
- Line 26: Concerning the OCT, you should use first the complete name before to use the abbreviation: optical coherence tomography (OCT)
- Line 44: “In vivo” should be written in Italic
- Line 142: Please give the name of the balsam used for the microscopic evaluation
- Line 194: The number “four” should be written in numbers “4” and not in letters (as you did in the line 172)
Reviewer 2 Report
The manuscript entitled, "An assay system to evaluate riboflavin/UV-A corneal phototherapy efficacy in a porcine corneal organ culture model" focuses on the development of a scoring method to evaluate corneal stromal organization and compactness, using a "brightness" score. While there is a need to have automated scoring mechanisms to evaluate histologic data, the scientific approach taken in the described study has some challenges and limitations.
The premise of the study is to validate a new technique to automate analysis of corneal stroma compactness using a well-established cross linking technique. While this initial experimental setup is valid, the actual histologic assessment of the tissue requires more attention. In a normal formalin-fixed corneal tissue, there is visible clefting of the stroma when stained with H&E and viewed using light microscopy. The absence of these clefts are actually used to determine the presence of disease (corneal edema, keratomalacia, cellular infiltration). Without properly establishing proper criteria for the healthy tissue, the remaining analysis is placed into question. In addition, the histologic images are of poor quality.
Using the mean and SD from Table 1, and redoing the ANOVA, there is no statistical significance between groups. There may be a trend with the injured corneas to have increased brightness, but it is not significant from my calculations. The remaining figures and tables further examine this data when it is broken down into regions of interest. It is also unclear the significance of these more focal analyses. The anterior stroma of mammals has a very different collagen arrangement when compared with the posterior cornea, and the tissue compliance is also very different, with the anterior stroma having a higher level of intertwining and stiffer ECM. Therefore, the brightness and all other interpretation is complicated by this underlying fact.
Lastly, a better utilization of the figures would be to demonstrate the concept of determining brightness of the stroma with images from the green channel in the manuscript and highlighting the areas of interest.
If one were to redesign this study, it would be best to use other modalities to validate the collagen stromal rearrangement which could then correlate with changes in "brightness".